# Unveiling the Maze: Branched-Chain Amino Acids Fueling the Dynamics of Cancer Metabolism and Progression

**DOI:** 10.3390/cancers17111751

**Published:** 2025-05-23

**Authors:** Kai Xu, Pratham Shah, Dhruvi Makhanasa, Md. Wasim Khan

**Affiliations:** Division of Endocrinology, Diabetes and Metabolism, Department of Medicine, University of Illinois at Chicago, Chicago, IL 60612, USA; kxu12@uic.edu (K.X.); pshah249@uic.edu (P.S.); drdhruvi0105@gmail.com (D.M.)

**Keywords:** BCAAs, metabolic reprogramming, cancer metabolism

## Abstract

Branched-chain amino acids (BCAAs) are essential nutrients that help the body build proteins and signal cells when to grow and divide. Recent research has found that cancer cells use BCAAs in special ways to support their growth and survival. BCAAs provide fuel and help cancer cells change how they use energy and nutrients, making tumors more aggressive. They influence key cell pathways that control metabolism and growth. This article reviews new findings on how BCAAs impact cancer cell behavior, showing that these amino acids play a dual role as building blocks and energy sources. It also explores how BCAAs interact with other systems in the body, potentially helping tumors resist treatment. Scientists believe that by better understanding how tumors depend on BCAAs, they can develop new cancer therapies that block or disrupt these processes. Targeting BCAA use in cancer cells could open the door to more effective treatments.

## 1. Introduction

Cancer remains a leading cause of mortality worldwide, in part due to the metabolic plasticity that enables tumor cells to adapt to nutrient stress and sustain uncontrolled proliferation. To ensure rapid growth, cancer cells must rewire their metabolism to meet increased energy and biosynthetic demands. In addition to adapting to nutrient limitations within the tumor microenvironment, cancer cells use metabolic intermediates as potent signaling molecules that promote tumor development. This ability to reprogram metabolism supports proliferation and drives oncogenic signaling. Metabolic dependencies, therefore, represent attractive therapeutic vulnerabilities. Among these, branched-chain amino acids (BCAAs)—leucine, isoleucine, and valine—have emerged as critical players in both normal physiology and malignancy, acting not only as building blocks for protein synthesis but also as regulators of signaling and energy metabolism [1]. BCAAs cannot be synthesized de novo in mammalian cells and must be obtained from the diet, which is why they are classified as essential amino acids. Over the past decades, studies have documented altered BCAA levels—both within cells and systemically—in cancer compared to normal tissues.

Under homeostatic conditions, BCAAs are imported into cells primarily via the L-type amino acid transporters LAT1 or LAT2 in complex with SLC3A2 and then undergo transamination by cytosolic BCAT1 or mitochondrial BCAT2 to form branched-chain keto acids (BCKAs). Subsequent irreversible decarboxylation by the branched-chain α-keto acid dehydrogenase (BCKDH) complex yields acyl-CoA derivatives that fuel the tricarboxylic acid (TCA) cycle [1]. This tightly regulated pathway provides both anabolic substrates and allosteric signals—most notably via the leucine-mediated activation of mTORC1—to balance protein synthesis, redox homeostasis, and energy production.

Tumor cells frequently hijack this system by upregulating BCAA transporters, stabilizing BCAT isoforms, and modulating BCKDH activity, thereby reprogramming BCAA catabolism to support biosynthesis, bioenergetics, and oncogenic signaling. In this review, we summarize how various cancers alter BCAA metabolism. It covers changes in gene expression, shifts in metabolite profiles, and effects on related metabolic pathways. Importantly, we explore potential therapeutic targets within the BCAA metabolism pathway, including currently available inhibitors. This review highlights new avenues for therapeutic intervention in cancer treatment by elucidating these metabolic changes.

## 2. BCAA Metabolism

BCAA metabolism is a tightly regulated, multi-step process that balances nutrient utilization with energy production and biosynthesis. In mammalian cells, this pathway begins with the cellular uptake of BCAAs, followed by transamination to BCKAs, and culminates in their irreversible oxidation within mitochondria. Each step involves distinct transporters and enzymes that together coordinate BCAA availability with metabolic demands.

### 2.1. Cellular Uptake

BCAAs enter mammalian cells predominantly via the L-type amino acid transporters LAT1 or LAT2 in a heterodimeric complex with SLC3A2. LAT1 (SLC7A5/SLC3A2) is broadly upregulated in rapidly proliferating tissues and many cancers, whereas LAT2 (SLC7A8/SLC3A2) exhibits wider expression in normal epithelia [2] (Figure 1). This transporter pairing ensures an efficient influx of leucine, isoleucine, and valine across the plasma membrane, coupling extracellular availability to intracellular metabolic programs.

### 2.2. Transamination

Once inside the cell, BCAAs undergo reversible transamination catalyzed by branched-chain aminotransferases: cytosolic BCAT1 and mitochondrial BCAT2 (Figure 1). BCAT1 is enriched in the brain, kidney, and ovarian tissues, whereas BCAT2 is ubiquitous except in the liver [2]. During this reaction, each BCAA donates its amino group to α-ketoglutarate, forming glutamate and the corresponding branched-chain keto acid (BCKA)—α-keto-β-methylvalerate from isoleucine, α-ketoisovalerate from valine, and α-ketoisocaproate from leucine. To access mitochondrial BCAT2, BCAAs cross the inner membrane via the SLC25A44 carrier [3]. Under physiological conditions, skeletal muscle serves as the primary site of BCAA transamination [4].

### 2.3. Oxidative Decarboxylation

The committed, irreversible step in BCAA catabolism occurs in mitochondria through the branched-chain α-keto acid dehydrogenase (BCKDH) complex (Figure 1). Composed of E1 (BCKDHA/BCKDHB), E2 (DBT), and E3 (DLD) subunits [5], BCKDH catalyzes the decarboxylation of BCKAs to acyl-CoA derivatives—acetyl-CoA from leucine and succinyl-CoA from isoleucine/valine—that feed into the tricarboxylic acid (TCA) cycle. The regulation of BCKDH is achieved via phosphorylation: PPM1K (PP2Cm) activates the complex by dephosphorylating the E1 subunit, while BCKDK inactivates it via E1 phosphorylation [6]. Downstream, propionyl-CoA carboxylase and methylmalonyl-CoA mutase convert these acyl-CoAs into succinyl-CoA, integrating them into gluconeogenesis, ketogenesis, and fatty acid synthesis pathways [4]. Notably, the mechanism by which cytosolic BCKAs are shuttled into mitochondria for oxidation remains to be elucidated.

## 3. Reprogramming BCAA Metabolism in Cancer

BCAAs serve multiple pro-tumorigenic functions by supplying essential building blocks, activating key growth pathways, and reprogramming cancer cell metabolism to adapt to the fluctuating tumor microenvironment. Cancer cells, which need extra carbon and nitrogen for rapid growth, biosynthesis, and energy, often rewire their BCAA breakdown pathways to meet these demands. This reprogramming can involve changes in enzyme gene expression, improvements in reaction efficiency, stabilization of key enzymes, and shifts in enzyme–substrate specificity.

The following are the principal mechanisms by which BCAA availability and catabolism facilitate cancer progression:*Provision of Carbon and Nitrogen for Biosynthesis*BCAAs supply both the carbon skeletons and amino groups required for de novo protein synthesis, supporting the rapid proliferation of malignant cells.


*Activation of mTORC1 Signaling*
Leucine acts as a direct allosteric activator of mTORC1, thereby promoting anabolic programs, cell growth, and survival under nutrient-replete conditions.


*Anaplerotic Support and Energy Generation*
Through BCAT-mediated transamination and BCKDH-driven oxidative decarboxylation, BCAAs generate acetyl-CoA and succinyl-CoA, replenishing TCA cycle intermediates and fueling ATP production.


*Maintenance of Redox Homeostasis*
Transamination of BCAAs yields glutamate, which is a precursor for glutathione synthesis; elevated glutathione buffers reactive oxygen species and enhances stress resistance.


*Augmentation of Nucleotide and Nonessential Amino Acid Synthesis*
BCAA-derived α-ketoglutarate and glutamate serve as substrates for the synthesis of nonessential amino acids and nucleotides, thus supporting DNA replication and repair.


*Epigenetic Regulation via α-Ketoglutarate*
Fluctuations in intracellular α-ketoglutarate concentrations modulate the activity of dioxygenase enzymes (e.g., TET, JmjC histone demethylases), altering gene expression profiles in favor of malignancy.


*Metabolic Plasticity and Adaptation*
By toggling between glycolytic and oxidative pathways in response to microenvironmental cues, cancer cells exploit BCAA catabolism to maintain bioenergetic flexibility.


*Modulation of the Immune Microenvironment*
Elevated extracellular BCAAs and their metabolites can impair T-cell activation and skew macrophage polarization, thereby fostering an immunosuppressive tumor niche.


*Cross-Talk with Lipid Metabolism*
BCAA-derived citrate and acetyl-CoA influence de novo lipogenesis and fatty acid oxidation, supporting membrane biogenesis and lipid-mediated signaling networks.

Below, we discuss these aspects of altered BCAA metabolism in specific cancer types—pancreatic ductal adenocarcinomas (PDAC), hepatocellular carcinoma (HCC), breast cancer, non-small cell lung cancer (NSCLC), leukemia, glioblastoma, and other cell types in the tumor microenvironment.

### 3.1. PDAC

In PDAC, intracellular BCAA levels are controlled by the BCAA transporter and are one of the essential aspects of determining BCAA metabolism in PDAC pathogenesis. Multiple studies have observed an elevated BCAA intake with an increased expression of BCAA transporters [3,4]. Without upregulation of BCAA transporters, BCAA metabolism is unlikely to contribute significantly to the development or progression of PDAC [5].

BCAT2, the first enzyme in the BCAA catabolic pathway, is found to be overexpressed at both the *mRNA* and protein levels in PDAC [3,4,5,6]. Ablation or inhibition of *BCAT2* reduces PDAC cell proliferation [6]. Recent studies have also explored BCAT2 protein stability in PDAC. Li et al. identified that USP1 deubiquitylates BCAT2 at K229 under high BCAA levels, and inhibiting this process leads to a slowdown of the PDAC colony formation [6]. Lei et al. discovered that acetylation at K44 leads to BCAT2 degradation by the ubiquitin–proteasome pathway under BCAA-deprived conditions, which efficiently inhibits PDAC proliferation [7]. This evidence puts *BCAT2* under the spotlight in PDAC pathogenesis.

Following the initial BCAT-mediated transamination step, the commitment to BCAA metabolism is determined by the activity of the BCKDH complex, which irreversibly decarboxylates BCKAs. Investigating how the expression of enzymes involved in this step is regulated—and how these changes contribute to the development of pancreatic ductal adenocarcinoma (PDAC)—is essential for understanding the metabolic dependencies of this cancer. Knocking down *BCKDHA*, a subunit of the *BCKDH* complex, in PDAC cells also substantially reduces proliferation, suggesting the importance of committing to BCAA metabolism in PDAC pathogenesis [3]. Similarly, Zhu et al. showed that BCKAs, produced by cancer-associated fibroblasts, are the preferred substrate for PDAC cells [8]. These studies suggest that both BCAAs and their oxidative catabolism play a significant role in PDAC progression.

### 3.2. HCC

The exploration of the role of BCAAs in HCC started in the early 2000s. In these studies, the authors have shown that BCAA supplementation benefits HCC patients or rodents receiving radiofrequency ablation [9,10,11]. In vitro studies have also demonstrated that BCAA supplementation leads to multiple mechanisms that mediate autophagy and apoptosis [12,13]. However, the results from recent studies say otherwise. Clinical studies have revealed that BCAA supplementation elevates serum albumin levels and supports remaining liver function [14]. Furthermore, a more comprehensive meta-analysis incorporating data from multiple cohorts across existing studies found that BCAA supplementation did not provide a protective effect against HCC [15,16]. However, these studies have not focused on the effect of BCAA metabolism itself on HCC pathogenesis. In preclinical studies, the findings are inconsistent, with varying results regarding the effects of BCAA supplementation or metabolism on disease progression. Kim et al. showed that *LAT1* is the most significantly upregulated amino acid transporter in HCC [17]. Genetic deletion of *LAT1* resulted in reduced liver tumor growth and weakened mTORC1 signaling, highlighting its critical role in supporting HCC progression [17]. The study by Ericksen et al. showed significant overexpression of *BCAT1* and *BCAT2* in HCC tumors compared to normal tissue. However, other enzymes in BCAA metabolism are either downregulated or have reduced activity (e.g., *BCKDH*), which leads to the accumulation of BCAAs and the subsequent activation of mTORC1 by binding to its regulatory complex, promoting cell growth, protein synthesis, and metabolism. Interestingly, reducing BCAA supplementation and increasing BCAA metabolism were associated with better outcomes in this study [18]. Conversely, BCAA catabolism is active and correlated with increased tumorigenesis through PPM1K-mediated phosphorylation of BCKDH under glutamine starvation [19].

In the current literature, BCAA metabolism impacts HCC progression in a nutrient-dependent manner, possibly due to the complex etiology that leads to the distinct metabolism of HCC, especially in conditions like glutamine starvation. Nevertheless, most evidence under normal glutamine concentration showed increased BCAA metabolism contributing to HCC progression. Further studies with more stringent categorization in clinical studies and different models of HCC in mimicking different metabolic phenotypes of HCC would facilitate deciphering nutrient-dependent BCAA metabolism in HCC.

### 3.3. Breast Cancer

Multiple studies have demonstrated that breast cancer has high BCAA catabolism at the reversible step controlled by BCAT [20,21,22]. *BCAT1* overexpression is a commonly reported finding in the literature [20,22,23]. Inhibiting BCAT1 by Eupalinolide B or the anti-cancer drug doxycycline reduces intracellular BCAA or BCKA levels, respectively [20,23]. Sadly, direct evidence using labeled BCAAs to trace the fate of BCAAs in breast cancer is lacking. With the inhibition of BCAT1 and doxycycline usage, the growth of breast cancer cells is inhibited [20,23]. It should be noted that *BCKDK* has been associated with cancer risks and theoretically negatively regulates BCAA metabolism [20,24,25]. First, *BCKDK* is negatively associated with a low recurrence-free survival rate [25], and genetic ablation or inhibition of *BCKDK* synergizes the cell cycle arrest effects of paclitaxel and increases cell death in breast cancer cells [20,24]. The effect of BCKDK on BCAA metabolism has been explored less, with no metabolomic data available. From this evidence, *BCAT1* overexpression leads to breast cancer progression. Regarding BCKA oxidation, direct evidence is still needed, such as measuring the BCKA oxidation metabolites to clarify how much *BCKDK* expression contributes to BCAA metabolism and, subsequently, breast cancer progression.

### 3.4. NSCLC

Preclinical and clinical data have shown increased intracellular BCAA levels in NSCLC [5,26,27,28]. This observation results from an adaptive change in BCAA metabolism in NSCLC, including increased *LAT1*, *BCAT1*, and *BCAT2* [5,27,29]. Importantly, in metastatic NSCLC cells in the brain, *BCAT1* expression is negatively associated with the survival rate [27]. A discrepancy exists regarding BCAA levels when knocking down *BCAT1* in NSCLC or brain metastasized NSCLC cells, with Zhang et al. showing no change and Mao et al. showing an increase in BCAA levels [27,29]. This may be explained by the malignancy level associated with *BCAT1* expression [27]. Like breast cancer, increased *BCKDK* and reduced *BCKDHA* further retain the BCAAs in NSCLC cells [26,28]. Genetic deletion of *BCKDK* increases apoptosis and reduces proliferation [26,28]. Interestingly, in NSCLC cells, BCKDK-induced citrate depletion was associated with increased ATP-citrate lyase activity, which converts citrate into acetyl-CoA and oxaloacetate. This raises the question of whether elevated BCKDK also promotes the reamination of BCKAs through the conversion of oxaloacetate to aspartate. However, direct evidence supporting this possibility is currently lacking [26].

### 3.5. Leukemia

It is widely accepted that intracellular BCAA levels are elevated in leukemia [30,31,32]. However, conflicting results exist regarding the flow of BCAAs. In acute myeloid leukemia (AML) cells of *Ezh2*^−/−^ *NRas*^G12D^ mice, the overexpression of *BCAT1* is responsible for converting BCKAs back to BCAAs [33]. Kikushige et al. showed that AML cells commit BCAAs to irreversible steps of BCAA metabolism and channel the final products to the TCA cycle [32]. Nevertheless, the literature agrees that *BCAT1* is overexpressed in AML cells [30,32,33,34,35,36,37,38]. Studies have shown multiple ways in which leukemia cells maintain abnormal elevation of *BCAT1* expression to meet their needs. Kikushige et al. demonstrated that PRC2, an important epigenetic regulator for AML stemness, is responsible for aberrant *BCAT1* expression and reprogrammed BCAA metabolism [32]. Loss of EZH2 in the PRC2 complex also drives *BCAT1* overexpression [33]. Meanwhile, N^6^-methyladenosine METTL16 and MSI2 can stabilize *BCAT1* mRNA to maintain BCAT1 translation in AML cells [37,38]. Notably, *BCAT1* expression has been associated with the outcome of leukemia [35,38]. Studies have shown that knockdown or inhibition of *BCAT1* leads to reduced survival and proliferation [34,36,39,40]. Limited studies have explored the expression of other enzymes and their activity in BCAA metabolism in leukemia. Liu et al. demonstrated that PPM1K is critical for BCAA metabolism [31]. However, no metabolites after the committed step of BCAA metabolism were quantified in this study [31]. Further studies are needed to determine whether intracellular BCAA levels, BCAA metabolism, or a combination of both are essential for leukemia progression.

### 3.6. Glioblastoma

Most studies report enhanced BCAA metabolism in glioblastoma [41,42,43], with a particular focus on the reversible transamination step mediated by BCAT enzymes. This interest is largely driven by the frequent overexpression of BCAT1 observed in glioblastoma, highlighting its potential role in tumor metabolism and progression [41,42,43]. For example, HIF1a/2a is important for maintaining *BCAT1* expression [42] in hypoxic conditions, and its deletion decreases *BCAT1* expression [42]. Studies have demonstrated that α-ketoglutarate, an important metabolite, can modulate DNA methylation, and its level regulates *BCAT1* expression [41,44]. Intriguingly, BCKDK, originally the kinase that phosphorylates BCKDH to reduce its activity, has been shown to prevent BCAT1 protein degradation by phosphorylation on its S5, S9, and T312 sites [45]. Like all other cancers, deletion, inhibition, or increased degradation of BCAT1 increases apoptosis and decreases proliferation [41,42,43,44,45]. Less research has been conducted on the latter step of BCAA metabolism in glioblastoma. Suh et al. observed increased radio label isotope incorporation into lactate and glutamate in glioblastoma from rats perfused with U-^13^C leucine, suggesting the commitment of BCAAs into later irreversible steps [46]. There is indirect evidence that knocking down the gene responsible for leucine oxidation, *HMGCL,* leads to cell death [47], suggesting that the oxidation of BCKAs is also responsible for glioblastoma pathogenesis. As in other cancers, *BCKDK* is linked to glioblastoma progression, with its knockout resulting in reduced tumor cell proliferation [47]. However, elevated BCKDK levels suppress BCKA oxidation, which appears to contradict the findings of increased isotope-labeled BCKA incorporation in glioblastoma. This discrepancy highlights the need for further studies to clarify the extent and regulation of BCKA oxidation in glioblastoma.

### 3.7. Other Cancers

Though the literature is less abundant, BCAA metabolism plays a role in gastric cancer and melanoma tumorigenesis. In gastric cancer, a mutation of BCAT1^E61A^ was identified in gastric cancer patients; this mutation leads to significantly elevated catalytic activity from BCAAs to BCKAs, while the ability to catalyze BCKAs to BCAAs remains similar, suggesting increased BCAA metabolism [48]. Such increased BCAA metabolism eventually leads to enhanced tumorigenesis [48]. In melanoma, both BCAT2 and BCKDHA are overexpressed. Inhibiting or suppressing either of these enzymes reduces colony formation and cell proliferation while promoting apoptosis [49,50].

## 4. Interplay Between BCAA Metabolism and Other Metabolic Networks

Growing evidence from the literature indicates that BCAA metabolism influences a range of other metabolic pathways, including those involved in glucose metabolism, non-essential amino acid synthesis, nucleotide biosynthesis, and fatty acid synthesis and breakdown. There is also growing evidence that BCAAs modulate the tumor microenvironment, which comprises multiple cell types, including immune cells, fibroblasts, and others [51]. These cells can also respond to elevated BCAA levels within the tumor, potentially influencing cancer progression. Additionally, reprogrammed BCAA catabolism in cancer cells may alter the immune landscape of the tumor microenvironment, further impacting tumor growth and immune responses. As previously mentioned, BCAA levels are increased in NSCLC, and T cells can take advantage of these elevated BCAA levels and increase the CD8^+^ population, leading to higher anti-tumor immunity [52]. Macrophages in glioblastoma can uptake the BCKAs from their environment and subsequently re-aminate to BCAAs, suppressing phagocytosis [53].

In this section, we discuss how BCAA metabolism modulates other metabolic pathways and discuss the implications for cancer progression.

### 4.1. Glucose Metabolism

The current literature suggests that increased BCAA metabolism in various cancers is generally associated with enhanced glycolysis, mitochondrial respiration, or both (Figure 2). The specific metabolic outcome appears to depend on the cancer type and its metabolic context [4,20,22,26,29,31,32,37,43,54]. In glioblastoma, Lu et al. demonstrated that the inhibition of BCAT1 disrupts mitochondrial membrane potential, leading to a reduction in oxidative phosphorylation, as shown by metabolomic analysis [43]. Similarly, knocking down BCKDK results in a marked decrease in several intermediates of the TCA cycle and glycolysis, further supporting the role of BCAA metabolism in maintaining central carbon metabolism in glioblastoma [54]. In leukemia, knocking down *PPM1K*, which is essential for reprogrammed BCAA metabolism, leads to significantly reduced glycolysis and oxidative phosphorylation capacity [31]. Han et al. also showed that BCAA metabolism contributes to increased TCA cycle metabolites and oxidative phosphorylation [32,37]. Knocking down N^6^-methyladenosine (m^6^A) *METTL16*, required for increased BCAA metabolism, leads to significantly reduced enrichment of isotopes in citrate and malate [8,37]. In breast cancer, while direct evidence linking BCAA metabolism to TCA cycle metabolite levels is limited, inhibition of key enzymes such as BCAT1 and BCKDK has been shown to reduce both basal and maximal respiration, as well as ATP production, indicating a functional role for BCAA metabolism in supporting mitochondrial energy output [20,22,23]. In NSCLC, proteomic analyses of TKI-resistant cells reveal an enrichment of glycolytic pathway proteins [29]. Both TKI-resistant cells and those overexpressing *BCKDK* show increased maximum glycolytic capacity [26,29]. However, there is a discrepancy in the reported basal glycolytic levels between studies [32,35], highlighting the need for further investigation. In PDAC, increased maximum glycolytic capacity and basal and maximum phosphorylation have been observed in cells overexpressing *BCAT2* [4]. Enhanced integration of BCAA metabolites into the TCA cycle by isotope tracing has also been observed [4,8,37].

The mechanism of glucose metabolism modification by reprogrammed BCAA metabolism is less explored. One of the mechanisms for changes in glycolysis can be due to increased α-ketoglutarate levels by BCAA metabolism, which consequently leads to demethylation of histone H3 lysine 27 on glycolytic genes and increases glycolytic activity in NSCLC [29]. Enhanced oxidative phosphorylation is likely due to the direct contribution of metabolites from BCAA metabolism to the TCA cycle [8,37]. Further studies are required to understand the link between glucose metabolism and BCAA metabolism.

### 4.2. Nonessential Amino Acids and Nucleotide Synthesis

Nonessential amino acids and nucleotides are essential building blocks for cancer cell proliferation. Reprogrammed BCAA metabolism significantly increases the production of glutamate, a byproduct of BCAA transamination, which in turn fuels the synthesis of other nonessential amino acids and nucleotides, supporting tumor growth and biosynthesis [4,5,19,37]. *BCAT2* overexpression in PDAC and *METTL16*-induced elevated BCAA metabolism in leukemia leads to the incorporation of N^15^ isotope from BCAAs into nonessential amino acids and nucleotides. This suggests that BCAAs contribute directly to the synthesis of these molecules, supporting cancer cell proliferation and metabolism [4,37]. A similar result was observed in HCC cells with glutamate starvation, but no difference was observed between normal and glucose starvation conditions [19]. This effect also holds in animal models of NSCLC, where N^15^-labeled BCAAs contribute to nonessential amino acid and nucleotide production [5]. Similarly, in gliomas with IDH1^R132H^ mutation, inhibition of BCAT transaminase activity by the oncometabolite (R)-2-hydroxyglutarate causes a metabolic shift. Isotope tracing shows that these cells reduce their reliance on BCAA metabolism for glutamate production and instead increase glutamate synthesis from glutamine [55].

### 4.3. Fatty Acid Metabolism

BCAA metabolism also plays a role in fat biosynthesis, but current findings are inconsistent due to limited available studies. BCAA starvation leads to the cellular accumulation of lipid droplets due to the failure to import fatty acids into the mitochondria in PDAC [56]. In HCC, reduced fatty acid oxidation, particularly the downregulation of *CPT1A* expression, leads to decreased acetyl-CoA production (Figure 2). This reduction in acetyl-CoA alters the epigenetic landscape by lowering acetylation levels, which in turn promotes an increase in BCAA metabolism [57]. Another study by Lee et al. demonstrated that knockdown of *BCKDHA* inhibits fatty acid biosynthesis. Interestingly, *CPT1A* is also downregulated in benzene-induced leukemia [58]. In melanoma, increased BCAA metabolism leads to higher expression of *FASN* and *ACLY* genes, suggesting higher de novo lipogenesis [49,50]. This observation introduces a new perspective on the interplay between BCAA metabolism and fatty acid oxidation.

The current literature suggests that altered BCAA metabolism boosts the production of nonessential amino acids and nucleotides by increasing glutamate levels from BCAAs. This, in turn, enhances glucose metabolism, although the extent of this effect varies across different cancer types. Research indicates that epigenetic modulation, driven by fluctuations in α-ketoglutarate levels, may cause cancer cells to become more dependent on glycolysis. Additionally, metabolites from BCAA metabolism likely influence changes in oxidative phosphorylation. The relationship between BCAA metabolism and fatty acid synthesis and oxidation is context-dependent, and further investigation is needed to fully understand its implications.

## 5. Therapeutic Strategies for Targeting BCAA Metabolism

Earlier, we discussed how BCAT and BCKDK expression in various cancers influences BCAA metabolism, promoting cancer growth by reprogramming metabolic pathways. Metabolites and genes from this catabolism further support cancer cell proliferation and survival. Current research is focused on identifying compounds that can regulate these enzymes to mitigate their pro-survival effects. This section reviews the latest preclinical and clinical compounds or molecules targeting BCAA metabolism.

### 5.1. BCAT Inhibitors

To date, several BCAT1 inhibitors have been developed, with gabapentin being the most widely used. Gabapentin, a leucine structural analog, inhibits BCAT1 without affecting BCAT2. Originally developed as a gamma-aminobutyric acid (GABA) analog for pain management, gabapentin effectively inhibits BCAT1 at millimolar concentrations. This selective inhibition makes it suitable for testing in cancers with *BCAT1* overexpression. Gabapentin has demonstrated its efficacy in preclinical models, where it inhibits glioblastoma colony formation and cell proliferation in vitro [59]. In NSCLC, gabapentin can reduce proliferation in *BCAT1* overexpressing TKI-resistant cells [50]. What is particularly intriguing is that the combination of gabapentin and tyrosine kinase inhibitors (TKIs), such as ASK120067 and osimertinib, reduced BCAT1 protein levels. This suggests that GABA analogs like gabapentin may not only inhibit BCAT1 directly but also potentially enhance the effects of TKIs, providing a promising therapeutic approach for targeting BCAT1 in cancer [50]. Novel GABA analogs that can overcome TKI resistance in NSCLC are being developed [60]. Excitingly, a clinical trial (NCT05664464) is ongoing, using a combination of gabapentin with a compound to inhibit glutamate secretion and receptor activation in conjunction with radiotherapy to explore the effectiveness of BCAT1 inhibition in glioblastoma patients [61]. Other compounds received less attention; nevertheless, they also showed exceptional preclinical results. Candesartan, an FDA-approved compound that was repurposed for the BCAT1^E61A^ mutation, was reported by Qian et al. In that paper, they proved the direct binding of candesartan with BCAT1^E61A^, with a reduction of gastric tumor proliferation both in vivo and in vitro [48]. Eupalinolide B is a compound from *Eupatorium lindleyanum* that inhibits BCAT1. Both in vitro and xenograft models show that eupalinolide B can slow the proliferation of triple-negative breast cancer [23]. IDH1^R132H^ mutation in glioma leads to overproduction of (R)-2-hydroxyglutarate, which, though not specifically, inhibits BCAT1 activity, leading to increased dependency on glutaminase [55]. Compared to BCAT1, BCAT2 inhibitors have received less attention. BCAT-IN-2 was first identified in a high-throughput screening in 2015; its effectiveness is proven by the retention of BCAAs in serum [62]. Later, it was used in multiple studies to reverse the metabolic dysfunction associated with fatty liver disease [63,64]. Telmisartan, a repurposed BCAT2 inhibitor, shows promising results in metabolic disease [65,66]. However, their efficacy in cancer prevention is lacking. A recent study in melanoma shows that BCAT-IN-2 leads to fewer colonies forming in vitro and reduces aberrant *FASN* and *ACLY* expression [50].

### 5.2. BCKDK Inhibitors

BCKDK, a kinase that inhibits BCKDH activity, has received attention in cancer research. Modulation of *BCKDK* expression shows promising results in preclinical studies. In 2023, a series of angiotensin II receptor blocker-like molecules were structurally identified as possible BCKDK inhibitors [67]. BT2 and valsartan have been shown to inhibit BCKDK in vitro [67], and BT2 also increases the sensitivity of anti-cancer drugs in preclinical breast and ovarian cancer models [24]. Recent studies reveal that enhanced BCAA metabolism boosts anti-tumorigenic capabilities of T-cells by increasing the population of CD8^+^ T-cells [52]. Similarly, BCKDK can enhance CAR-T-cell therapy efficacy [68], suggesting that more comprehensive studies on BCKDK inhibition regarding cancer and cancer immunity are necessary.

### 5.3. Other Potential Therapeutic Targets

In cancers with overexpression of BCAT1, studies have shown that intracellular BCAAs accumulate, accompanied by a reduction in the expression of downstream enzymes. This raises the question of whether blocking BCAA import or enhancing the activity of the BCKDH complex to metabolize excess BCAAs could slow cancer progression. However, blocking BCAA influx is not a practical option, as it also affects the transport of other amino acids in different cell types, particularly immune cells. Therefore, metabolizing excess BCAAs remains the primary strategy. To move forward, several questions need to be addressed: First, can BCKAs, converted by BCAT1, be transported into mitochondria for further metabolism? Second, can overexpression of BCAT2 lead to a sustained increase in BCAA metabolism? Third, is simply channeling BCAAs to mitochondria sufficient, or is increased downstream enzymatic activity also necessary? These questions highlight important areas for future research.

### 5.4. Limitations and Challenges

Despite promising preclinical data, several limitations temper the translation of BCAA-targeted therapies into clinical practice. First, many existing inhibitors—such as gabapentin for BCAT1—require millimolar concentrations to achieve effective enzyme blockade, raising concerns about off-target effects, systemic toxicity, and achievable dosing in patients. Moreover, the structural conservation between BCAT isoforms and related aminotransferases complicates the design of truly selective compounds, increasing the risk of unintended interference with normal amino acid homeostasis [59].

Second, redundancy and plasticity within amino acid transport and catabolic networks may undermine single-target approaches. For example, blocking LAT1 can lead to compensatory upregulation of LAT2 or alternative transporters, while enhancing BCKDH activity may simply redirect excess BCKAs into other anaplerotic or biosynthetic pathways [59]. Furthermore, most studies have been performed on cell lines or xenografts that lack the complexity of the human tumor microenvironment—particularly stromal and immune components—which can dramatically influence drug penetration, metabolic cross-talk, and therapeutic response.

Finally, clinical evidence is still sparse [61]. Few trials have directly evaluated BCAA-modulating agents in oncology, and biomarker strategies for patient selection (e.g., BCAT1/2 expression levels, circulating BCAA profiles) remain underdeveloped. Moving forward, combination regimens that pair BCAA metabolism inhibitors with established chemotherapies, targeted agents, or immunotherapies may be necessary to overcome adaptive resistance. Rigorous pharmacokinetic, safety, and efficacy studies in well-characterized patient cohorts will be critical to validating BCAA metabolism as a viable therapeutic axis.

## 6. Concluding Remarks

Cancer cells commonly upregulate BCAA metabolism through increased BCAT and BCKDK activity—driven by enhanced transcription, mRNA stability, and protein stability—to fuel their anabolic needs. The relative contribution of BCAT1 versus BCAT2 dictates whether BCAAs undergo reamination or oxidation, and additional enzymes in this network remain underexplored. By generating metabolic intermediates that feed into glucose, nucleotide, amino acid, and fatty acid pathways, or by modulating epigenetic and transcriptional programs, BCAA catabolism reshapes central carbon and nitrogen fluxes in tumors. Although targeting BCAT and BCKDK offers a promising therapeutic avenue, the development of potent, selective inhibitors and rigorous clinical evaluation are still needed. In this review, we have detailed how cancer cells rewire BCAA metabolism at every step—from transporter upregulation and BCAT-mediated transamination to dysregulated BCKDH activity—to support anabolic growth, redox homeostasis, and oncogenic signaling, highlighting both tumor-specific adaptations and shared mechanistic themes, and surveying emerging strategies for intervention.

### 6.1. Outstanding Questions

Mitochondrial Trafficking of BCKAs: What are the transport mechanisms and carriers responsible for shuttling BCAT-derived keto acids into the mitochondria for oxidation?Isoform-Specific Contributions: How does BCAT1 versus BCAT2 overexpression differentially affect tumor metabolism and growth in distinct cancer contexts?Context-Dependent Effects: Under which nutrient or microenvironmental conditions (e.g., glutamine deprivation, hypoxia) does BCAA catabolism most critically drive tumor progression?Immune and Stromal Interactions: How does tumor-intrinsic BCAA metabolism reshape the function of tumor-infiltrating immune cells and cancer-associated fibroblasts?Biomarker Development: Which metabolic or genetic readouts (plasma BCAA levels, BCAT/BCKDK expression) best predict responsiveness to BCAA-targeted therapies?

### 6.2. Future Directions

Looking ahead, advancing our understanding of BCAA metabolism in cancer will require integrated, multidisciplinary approaches. First, applying targeted metabolomics in genetically engineered mouse models and patient-derived xenografts will allow precise mapping of BCAA-modulated carbon and nitrogen fluxes within the physiologically relevant tumor microenvironment. Complementing these in vivo studies, high-throughput CRISPR/Cas9 knockout and activation screens can identify new regulators of BCAA transport and catabolism, revealing additional therapeutic targets. At the same time, single-cell metabolomic technologies promise to uncover intratumoral heterogeneity in BCAA pathway activity across both malignant cells and stromal elements, shedding light on cell-specific dependencies. On the translational front, rationally designed combination regimens—pairing BCAA metabolism inhibitors with standard chemotherapies, targeted kinase inhibitors, or immunotherapies—should be evaluated to overcome adaptive resistance. Finally, leveraging structure-based drug design and phenotypic screening efforts will be critical for generating potent, isoform-specific BCAT and BCKDK inhibitors with the pharmacokinetic and safety profiles needed for clinical testing.

## Figures and Tables

**Figure 1 cancers-17-01751-f001:**
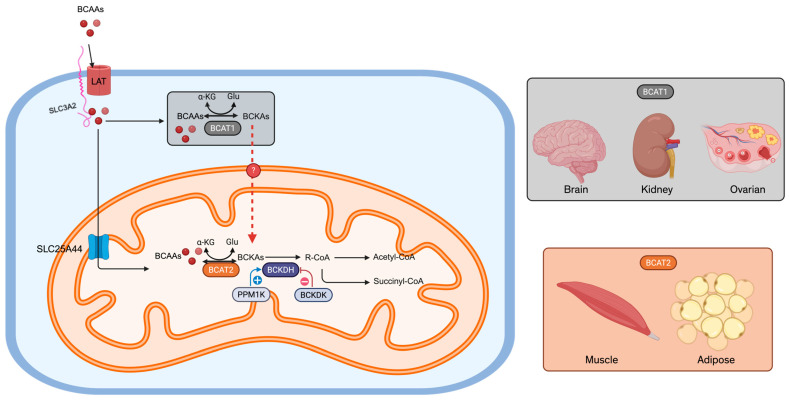
Systematic BCAA metabolism. BCAAs enter muscle cells through a channel made of LAT (LAT1 or LAT2) and SLC3A2 or from protein breakdown. In the cytosol, BCAAs donate the amine group to α-ketoglutarate to form glutamate and BCKAs by the enzyme BCAT1 (expressed in brain, kidney, and ovarian tissues) in a reversible reaction. BCAAs are transported into mitochondria through SLC25A44. BCAT2 (ubiquitously expressed, especially in muscle and adipose tissue) carries out the same transamination and is a reversible reaction. Then, it converts BCKAs enters circulation and is converted into downstream metabolites in the commitment step by BCKDH in hepatocyte mitochondria. BCKDH activity is dictated by its phosphorylation status, with dephosphorylation controlled by PPM1K, leading to activation, whereas phosphorylation is controlled by BCKDK, leading to inactivation. Several steps are omitted here. The final products of BCAA metabolism are acetyl-CoA and succinyl-CoA. The question remains as to how BCKAs enter mitochondria (denoted by a red dotted line with a question mark). BCAAs: branched-chain amino acids; BCKAs: branched-chain keto acids; α-KG: α-ketoglutarate; Glu: glutamate. Created in bioRender.

**Figure 2 cancers-17-01751-f002:**
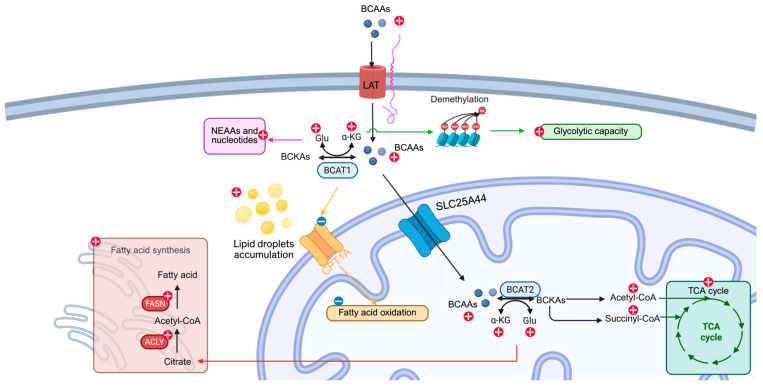
Metabolic pathways affected by BCAA metabolism. Elevated BCAA metabolism contributes to increased glycolytic capacity. One mechanism is through α-ketoglutarate-mediated demethylation (light green line and text box), non-essential amino acid and nucleotides synthesis (magenta line and text box), TCA cycle input (dark green line and text box), fatty acid synthesis (dark red line and text box), and decreased fatty acid oxidation (yellow line and text box). BCAAs: branched-chain amino acids; BCKAs: branched-chain keto acids, α-KG: α-ketoglutarate; Glu: glutamate; NEAA: non-essential amino acid; TCA cycle: tricarboxylic acid cycle. Created using bioRender.

## Data Availability

Not applicable.

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
