# Peer review of "Unveiling the Maze: Branched-Chain Amino Acids Fueling the Dynamics of Cancer Metabolism and Progression"

_cancers, 2025, doi:10.3390/cancers17111751_

Round 1
Reviewer 1 Report
Comments and Suggestions for Authors
I appreciated reading the review on the maze of branched chain amino acids in cancer. The authors did a decent job outlining the paper and ideas to reflect the importance of this subject. In terms of the science, I am satisfied with the ideas and presentation. However, I have some concerns regarding the readability of the paper. This may seem minor, nevertheless it requires attention and care. Many times, reading the manuscript sounded laborious and not easy to follow. It is definitely a job that a good editor can resolve. Moreover, the incessant use of abbreviations, especially when the full name was not previously mentioned clouded the reading experience.
There were some mistakes: for instance, line 51 on page 2: "During BCAT deamination" can be more accurately expressed "During BCAT transamination" because although there is a removal of the amine group from one amino acid, this group is transferred to the alpha keto acid. The enzymes are aminotransferases.
In the same line (51), a-ketoglutarate must be alpha (or α).
The gene names must be in italics.
In line 59 on page 2, "leading to its activation and deactivation" is misleading. It needs a comma and the word respectively, because as is it conveys the message that something causes both the activation and deactivation at the same time.
In line 63 on page 2 there is a missing reference (written as (Ref)).
In Figure 1: all CoAs must be written with a capital C rather than coA.
Oxidation is written with an "I" not with a "y"
In Figure 1 legend, there is a lack of identifying the abbreviations. The title of the figure is "BCAA influx into a cell by a heterodimer channel". This should be "through" because by denotes active involvement. Also, in the legend, there is a reference to a red dotted line with a question mark. I could not find it on the figure itself.
In section 3.1 (line 92 page 3), there seems to be statements of multiple conflicting results without synthesis from the authors. It would be nice to summarize the effects of BCAA on ubiquitination of certain proteins and their degradation in light with biological processes and functions.
Similar comments can be echoed for section 3.2 and 3.3. There needs to be authors' synthesis and summary of the multiple conflicting effects of BCAAs.
In section 3.3. what is the relationship between BCKDK inhibition and paclitaxel effects? This is not discussed.
In line 226 the authors decided to reuse "branched chain amino acids" as if it were introduced first in the paper.
Figure 2 legend is most likely switched with Figure 1 legend... notice the red arrow... Please double check.
Figure 2: Same comment as the one for CoA for Fig. 1.
Overall the manuscript is promising but it requires further editing and synthesis of major ideas in the words of the authors.
Comments on the Quality of English Language
All suggestions were presented previously.
Author Response
We thank the Editor and the Reviewers for a thorough evaluation of our manuscript and appreciate their valuable insights and suggestions. Below is our point-by-point response, with our answers to the concerns raised (in italics), and we have revised the manuscript accordingly. The changes in the manuscript are highlighted. Below are the answers to the concerns raised (in italics).
Reviewer 1
I appreciated reading the review on the maze of branched-chain amino acids in cancer. The authors did a decent job outlining the paper and ideas to reflect the importance of this subject. In terms of the science, I am satisfied with the ideas and presentation. However, I have some concerns regarding the readability of the paper. This may seem minor, nevertheless it requires attention and care. Many times, reading the manuscript sounded laborious and not easy to follow. It is definitely a job that a good editor can resolve. Moreover, the incessant use of abbreviations, especially when the full name was not previously mentioned clouded the reading experience.
Response: Thank you for your comments. We have made revisions throughout the manuscript to enhance clarity and readability.
There were some mistakes: for instance, line 51 on page 2: "During BCAT deamination" can be more accurately expressed "During BCAT transamination" because although there is a removal of the amine group from one amino acid, this group is transferred to the alpha keto acid. The enzymes are aminotransferases.
Response: Thank you for bringing this to our attention. We have made significant revisions to this paragraph.
In the same line (51), a-ketoglutarate must be alpha (or α).
Response: We appreciate your feedback and have corrected this on page 2, line 81.
The gene names must be in italics.
Response: We appreciate reviewers’ comments, we have italicized all the gene symbols.
In line 59 on page 2, "leading to its activation and deactivation" is misleading. It needs a comma and the word respectively, because as is, it conveys the message that something causes both the activation and deactivation at the same time.
Response: Thank you for bringing this to our attention. We have made the necessary corrections.
In line 63 on page 2 there is a missing reference (written as (Ref)).
Response: We have added this reference.
In Figure 1: all CoAs must be written with a capital C rather than coA.
Oxidation is written with an "I" not with a "y"
Response: We apologize for the error and have corrected all misspellings.
In Figure 1 legend, there is a lack of identifying the abbreviations. The title of the figure is "BCAA influx into a cell by a heterodimer channel". This should be "through" because by denotes active involvement. Also, in the legend, there is a reference to a red dotted line with a question mark. I could not find it on the figure itself.
Response: We apologize for any mistakes; we have corrected all the misspellings.
In section 3.1 (line 92 page 3), there seems to be statements of multiple conflicting results without synthesis from the authors. It would be nice to summarize the effects of BCAA on ubiquitination of certain proteins and their degradation in light with biological processes and functions.
Response: Thank you for pointing it out, we have rewritten the portion with clarification on page 5, lines 171 to 175.
Similar comments can be echoed for section 3.2 and 3.3. There needs to be authors' synthesis and summary of the multiple conflicting effects of BCAAs.
Response: Thank you for pointing it out, we have rewritten the portion with clarification on pages 5-6, lines 213 to 220 for section 3.2 and page 6, lines 233 to 237 for section 3.3.
In section 3.3. what is the relationship between BCKDK inhibition and paclitaxel effects? This is not discussed.
Response: Thank you for highlighting this detail; we have included it in section 3.3 on page 6, lines 230 to 232.
In line 226 the authors decided to reuse "branched chain amino acids" as if it were introduced first in the paper.
Response: All abbreviations have been corrected.
Figure 2 legend is most likely switched with Figure 1 legend... notice the red arrow... Please double check.
Response: We sincerely apologize for this mistake and have rectified it.
Figure 2: Same comment as the one for CoA for Fig. 1.
Response: We apologize for this mistake and have corrected it.
Overall the manuscript is promising but it requires further editing and synthesis of major ideas in the words of the authors.
Reviewer 2 Report
Comments and Suggestions for Authors
The authors have submitted an important manuscript regarding BCAA metabolism in multiple cancer types with additional discussion on the tumor microenvironment and therapeutic implications. The manuscript provides a concise overview of the topic and covers divergent reports in the literature helping to highlight context dependencies regarding the role of BCAAs in cancer metabolism.
Addressing the comments below will help to improve the final version of the manuscript.
1. The authors may consider including more detail in Section 2 regarding liver/skeletal muscle inter-organ relationships for BCAA metabolism. For example, the preferential catabolism/transamination in skeletal muscle, transport of BCKAs from skeletal muscle to liver, and the biochemical reasons for these types of relationships, etc.
2. Portions of the figure legends for Figures 1 and 2 are switched and need to be corrected.
3. While Figure 2 is used to highlight a few additional details (e.g., PPM1K, BCKDK), it is essentially a less detailed version of Figure 1 creating redundancy and contains a large portion of unused space. Creatively reworking Figure 2 will increase the quality of the manuscript for the reader.
Minor edits:
Line 63: reference needed
Lines 89-91: inconsistent capitalization of cancer types. Capitalization is not necessary.
Lines 150, 248, 366: use of retard/retarded should be modified to alternative terminology (e.g., inhibited, slowed), etc.
Line 262: “...another metabolic pathway.” should be specified.
Line 297: “...leads to demethylation...” needs better explanation/context.
Line 350: “...usage in BCAT1 to increase cancer.” is unclear.
Author Response
Reviewer 2
The authors have submitted an important manuscript regarding BCAA metabolism in multiple cancer types with additional discussion on the tumor microenvironment and therapeutic implications. The manuscript provides a concise overview of the topic and covers divergent reports in the literature helping to highlight context dependencies regarding the role of BCAAs in cancer metabolism.
Addressing the comments below will help to improve the final version of the manuscript.
- The authors may consider including more detail in Section 2 regarding liver/skeletal muscle inter-organ relationships for BCAA metabolism. For example, the preferential catabolism/transamination in skeletal muscle, transport of BCKAs from skeletal muscle to liver, and the biochemical reasons for these types of relationships, etc.
Response: We have added this information to the legend of Figure 1.
- Portions of the figure legends for Figures 1 and 2 are switched and need to be corrected.
Response: Our apologies, we have corrected this mistake.
- While Figure 2 is used to highlight a few additional details (e.g., PPM1K, BCKDK), it is essentially a less detailed version of Figure 1 creating redundancy and contains a large portion of unused space. Creatively reworking Figure 2 will increase the quality of the manuscript for the reader.
Response: We have redrawn the figure and updated the legend accordingly.
Minor edits:
Line 63: reference needed
Response: Thank you for pointing out, we have added this reference.
Lines 89-91: inconsistent capitalization of cancer types. Capitalization is not necessary.
Response: Thank you for bringing this to our attention. We have corrected the issue in lines 108 to 110.
Lines 150, 248, 366: use of retard/retarded should be modified to alternative terminology (e.g., inhibited, slowed), etc.
Response: We have replaced the terms "retard" and "retarded.”
Line 262: “...another metabolic pathway.” should be specified.
Response: We have specified it on page 7 from lines 309 to 312.
Line 297: “...leads to demethylation...” needs better explanation/context.
Response: We have specified it on page 9, line 362.
Line 350: “...usage in BCAT1 to increase cancer.” is unclear.
Response: We have corrected it on page 10, line 417.
Round 2
Reviewer 1 Report
Comments and Suggestions for Authors
The authors responded to the queries raised.